# Reduced Oxidative Susceptibility of Lp(a) and LDL Fractions as a Pleiotropic Effect of Lipoprotein Apheresis in Patients with Elevated Lp(a) and ASCVDs

**DOI:** 10.3390/ijms252413597

**Published:** 2024-12-19

**Authors:** Aleksandra Krzesińska, Joanna Marlęga-Linert, Gabriela Chyła-Danił, Marta Marcinkowska, Paulina Rogowska, Katarzyna Stumska, Marcin Fijałkowski, Marcin Gruchała, Maciej Jankowski, Agnieszka Mickiewicz, Agnieszka Kuchta

**Affiliations:** 1Department of Clinical Chemistry, Medical University of Gdańsk, 80-211 Gdańsk, Poland; aleksandra.krzesinska@gumed.edu.pl (A.K.); gabriela.chyla@gumed.edu.pl (G.C.-D.); p.rogowska0311@gumed.edu.pl (P.R.); kasiapot@gumed.edu.pl (K.S.); maciej.jankowski@gumed.edu.pl (M.J.); 21st Department of Cardiology, Medical University of Gdańsk, 80-211 Gdańsk, Poland; joanna.marlega-linert@gumed.edu.pl (J.M.-L.); marta.marcinkowska@gumed.edu.pl (M.M.); marcin.fijalkowski@gumed.edu.pl (M.F.); marcin.gruchala@gumed.edu.pl (M.G.); aga.mickiewicz77@gmail.com (A.M.)

**Keywords:** lipoprotein(a), oxidized lipids, atherosclerosis, apolipoproteins, LDL, oxidation, lipoprotein apheresis, autotaxin, cardiovascular diseases

## Abstract

Oxidative modifications of lipoproteins play a crucial role in the initiation of atherosclerotic cardiovascular diseases (ASCVDs). Nowadays, the one effective strategy for the treatment of patients with hyperlipoproteinemia(a) is lipoprotein apheresis (LA), which has a pleiotropic effect on reducing the risk of ASCVDs. The significance of oxidative susceptibility of the LDL fraction in ASCVDs has been extensively studied. Whether LA alters the susceptibility of lipoprotein(a) to oxidative modifications remains an unresolved issue. In this study, we isolated lipoprotein fractions by ultracentrifugation in patients with hyperlipoproteinemia(a) undergoing apheresis (LA group) at three time points and patients who were qualified for LA but did not consent to the procedure (non-LA group). We performed copper-mediated oxidation of Lp(a) and LDL fractions and determined autotaxin activity. After apheresis, we observed a lower susceptibility to oxidation of the Lp(a) and LDL fractions as expressed by the extended value of oxidation lag time, decreased slope of the oxidation curve, and decreased final concentration of conjugated dienes. No significant differences were found between these parameters before and 7 days after LA. Additionally, both patients undergoing and not undergoing LA had a significant correlation between autotaxin activity and all parameters characterizing susceptibility to oxidation in the Lp(a) fraction. Our results demonstrate that the pleiotropic effect of apheresis may be related to the reduced oxidative susceptibility of Lp(a) and LDL particles, which may influence the reduction in ASCVD risk in patients undergoing apheresis. The results of the rebound effect 7 days after LA will contribute to a better definition of apheresis frequency guidelines.

## 1. Introduction

Atherosclerotic cardiovascular diseases (ASCVDs) are a major health problem in Western countries and constitute the main cause of morbidity and mortality. Multiple investigations have confirmed that the key initiating event in atherogenesis is the retention of low-density lipoprotein (LDL) cholesterol (LDL-C) and other cholesterol-rich apolipoprotein B (apoB)-containing lipoproteins within the arterial wall [1,2,3,4]. However, the initiation of atherosclerosis occurs with the involvement of modified LDL particles, and the most common cause of these modifications is oxidative changes and the formation of oxidized LDL particles (ox-LDL) [5,6]. This is confirmed by the demonstrated in vivo presence of ox-LDL in atherosclerotic lesions and higher levels of ox-LDL in people with cardiovascular risk factors compared to healthy individuals [7,8].

In the last decade, particular interest among apoB-containing lipoproteins has focused on a LDL-like particle, lipoprotein(a) (Lp(a)), which differs from LDL by the presence of an additional protein, apolipoprotein(a) (apo(a)) [9]. Lp(a), due to a longer half-life than LDL, is even more exposed to oxidative modifications than LDL and thus accelerated uptake into macrophages, which are precursors of atherosclerotic foam cells [10]. The proatherogenic mechanism of Lp(a) action is multidirectional and probably also associated with its importance in binding and transporting proinflammatory oxidized phospholipids (oxPL) [11]. Oxidized Lp(a), highly enriched in oxPL, leads to increased oxidative stress and inflammation—common pathomechanisms of vascular endothelial dysfunction leading to atherosclerosis [12,13,14]. Additionally, autotaxin (ATX), an extracellular phospholipase D, can enhance the pathological mechanism of oxPL. ATX hydrolyzes lysophosphatidylcholine to form lysophosphatidic acid, which is a highly active metabolite and contributes to inflammation, fibrosis, and osteogenesis [15,16]. Bourgeois et al. concluded that ATX is preferentially transported by Lp(a) and may be a novel biomarker for CAVD. The association of ATX with the toxic effect of Lp(a) has been confirmed by studies that have indicated an independent association of levels of ATX transported by Lp(a) with the presence of aortic valve stenosis (AVS) [15,16]. In a study by Torzewski et al., immunochemistry of human aortic valve leaflet lesions from subjects with different grades of AVS showed a strong presence of ATX, particularly in valve leaflets with advanced stages of calcification [17]. Epidemiological and Mendelian randomization studies have shown that increased Lp(a) concentration is an independent causal risk factor for ASCVDs and calcific aortic valve disease (CAVD) [18,19,20]. It is established that the curve of Lp(a) concentration and ASCVD risk association by concentration of Lp(a) follows an almost linear fashion [9,21]. Standard lipid-lowering medications, such as statins and ezetimibe, are ineffective in reducing Lp(a) levels [22]. In the era of RNA-based therapies being investigated in clinical trials, a new insight is represented by the antisense oligonucleotides and silencing RNA drugs (pelacarsen, olpasiran, zerlasiran, lepodisiran), which are able to affect the synthesis of apo(a) and further secretion of Lp(a) in hepatocytes [23,24,25,26,27,28]. Another molecule, muvalaplin, inhibits Lp(a) formation by blocking the apo(a) formation [29,30]. However, today, the one effective and available strategy for the treatment of patients with high Lp(a) levels is lipoprotein apheresis (LA), an extracorporeal procedure removing apoB-containing particles, mainly LDL and Lp(a) [9,31,32]. Currently, the most common indication for LA is an elevated concentration of Lp(a) with progressive cardiovascular disease (CVD). LA lowers Lp(a) levels by 70–75% per session, translating into a reduction in the rate of cardiovascular (CV) events [33,34,35,36]. Published data showed that regularly performed LA in CVD patients with elevated Lp(a) concentrations and near-normal LDL-C results in a significant reduction in the rate of CV events and improves myocardial perfusion reserve and exercise capacity [37]. CV benefits of LA therapy seem to be associated with not only a lipid-lowering effect but also with additional pleiotropic effects [38,39]. One possible mechanism of LA pleiotropic action is the effect on the quality of lipoprotein fractions, including their susceptibility to oxidation, as well as the change in the composition of lipoprotein classes and the transport of proatherogenic proteins [10].

The aim of our study was to assess the effects of the LA procedure on the quality of lipoprotein fractions in patients with high Lp(a) concentrations undergoing regular biweekly LA treatment. Additionally, we analyzed the quality of lipoprotein fractions in patients who were qualified for LA treatment but did not consent to this procedure. Considering the clinical importance of oxidized forms of lipoproteins in the pathogenesis of atherosclerosis, our study may also help to explain the pleiotropic effect of LA associated with a reduction in CV events. In addition, the results 7 days after LA may be the basis to better define guidelines for the frequency of LA use in patients with high Lp(a) concentrations.

## 2. Results

The baseline clinical characteristics of the study population are shown in Table 1. Of the 24 patients undergoing LA treatment (LA group), 14 were diagnosed with hypertension, 4 were diagnosed with diabetes, and 11 declared smoking habits. In the group of patients with high Lp(a) levels but not undergoing LA (non-LA group), 12 were diagnosed with hypertension, 4 were diagnosed with diabetes, and 8 declared smoking habits.

Table 2 shows the values of biochemical parameters in both groups of patients. We observed no significant differences in basic lipid profile parameters between the individuals in the LA group and the non-LA group. However, patients not treated with LA were characterized by significantly higher apoE concentrations than patients undergoing regular biweekly LA. Compared to the non-LA group, subjects from the LA group had significantly decreased activity of ATX. The univariate correlation analysis indicated that ATX activity positively correlated with Lp(a) concentrations in both groups (LA group: R = 0.48, *p* = 0.031; non-LA group: R = 0.55, *p* = 0.042). No significant differences were observed in the 8-iso-PGF2α as well as in the apolipoprotein profile of the HDL fraction of both analyzed groups.

We analyzed the impact of a single LA procedure on changes in biochemical parameters and showed the dynamics of these changes. The values before LA, after LA, and 7 days after LA are reported in Table 3. As expected, the MONET apheresis technique was effective in lowering Lp(a) (by 62%) and LDL-C (by 48%) and resulted in a reduction in TC and TG concentrations (by 41% and 45%, respectively). The decrease in the HDL-C concentration was lower (by 13%) but still statistically significant. We also observed a significant 35% decrease in the phospholipids levels along with a significant 14% reduction in the ATX activity. No significant differences were found in 8-iso-PGF2α concentrations before, after, and 7 days after LA.

A single session of LA resulted in a reduction in apolipoprotein levels and lipid profile parameters (Table 3). The concentration of apoB was reduced by 51%, apoC-III by 47%, and apoE by 36%. ApoA-I concentration was significantly decreased by 19%. Additionally, we noticed a 43% reduction in the apoB/apoA-I ratio. All analyzed values of apolipoproteins and biochemical parameters returned to their baseline values 7 days after the LA procedure.

In parallel, we evaluated the apolipoprotein profile in the HDL fraction and showed a significant decrease in concentrations of apoA-I (by 10%), apoE (by 25%), and apoC-III (by 68%) after LA treatment. HDL apoA-I/apoE and apoA-I/apoC-III ratios increased significantly after the procedure. No significant differences were observed between these parameters before and 7 days after the LA session.

Parameters assessing lipoprotein oxidation susceptibility (lag time, the slope of the oxidation curve, and the increase in absorbance) changed significantly after the LA session for both the Lp(a) (Figure 1) and LDL fraction (Figure 2). As a result of the LA treatment, the value of the oxidation lag time extended significantly by 49% for the Lp(a) fraction and by 77% for the LDL fraction. Analysis of the slope of the oxidation curve showed a significant decrease in the Lp(a) and LDL fraction after the LA procedure (by 52% and 72%, respectively). We also observed that patients after the LA session had a significantly decreased final concentration of conjugated dienes, illustrated by the absorbance at 234 nm of 37% for the Lp(a) fraction and 45% for the LDL fraction. However, no significant differences were observed in any of the parameters assessing lipoprotein susceptibility to oxidation in subjects before and 7 days after LA treatment in both Lp(a) and LDL fractions.

Analysis of the average values of conjugated dienes in the Lp(a) and LDL fractions at each time point showed that these values differed significantly (*p* < 0.001) before and after the LA procedure (Figure 3).

As shown in Table 4, in the group of patients undergoing LA treatment, there was a significant correlation between serum Lp(a) concentration and the slope angle of the oxidation curve for the Lp(a) fraction (R = 0.47; *p* = 0.021). In this group of patients, there was no significant correlation between serum Lp(a) concentration, oxidation lag time, and the increase in absorbance in the Lp(a) fraction. However, the non-LA group showed a significant negative correlation only between Lp(a) concentration and oxidation lag time in the Lp(a) fraction (R = −0.55; *p* = 0.044). In addition, both study groups were characterized by a significant negative correlation between serum ATX activity and oxidation lag time in the Lp(a) fraction. In both the LA and non-LA groups, ATX activity also correlated significantly with the slope of the oxidation curve and the increase in absorbance at 234 nm in the Lp(a) fraction. Parallel to these results, in the LA group, we observed a significant correlation between serum LDL-C concentration and the slope angle of the oxidation curve of the LDL fraction (R = 0.59; *p* = 0.002) (Table 4).

No statistically significant differences in parameters describing susceptibility to oxidative modification of lipoprotein fractions were found between the study groups (Table 5). However, in the LA group, the oxidation lag time was extended and the slope of the oxidation curve was decreased more than in the non-LA group for both Lp(a) and LDL fractions. Analysis of the average values of conjugated dienes in the Lp(a) and LDL fractions at the each time point showed that these values differed significantly (*p* < 0.001) among patients undergoing and not undergoing regular biweekly LA procedure (Figure 4).

## 3. Discussion

Our study is the first to show the positive impact of the LA procedure on parameters characterizing the in vitro copper-mediated oxidation susceptibility of the Lp(a) fraction.

Oxidative stress is considered a major factor in the initiation and progression of ASCVDs. Increased susceptibility to oxidation has been proven in individuals at high risk of CVD and has been claimed to predict the occurrence of coronary heart disease (CHD), independent of traditional risk factors [7]. We found that all study parameters characterizing the oxidation susceptibility of lipoproteins (oxidation lag time, the slope of the oxidation curve, and the increase in absorbance at 234 nm) changed significantly as a result of the LA procedure (Figure 1 and Figure 2). The oxidation lag time was extended significantly by 49% for the Lp(a) fraction and by 77% for the LDL fraction, which indicates that lipoproteins started to oxidize after the LA procedure rather than before it. We showed also slower oxidation dynamics, expressed as the slope of the oxidation curve, which decreased significantly by 52% and 72% for Lp(a) and LDL fractions, respectively. These results were also confirmed in both fractions by the 40% decrease in the final concentration of conjugated dienes illustrated by the absorbance at 234 nm. Our results are in line with previous reports, demonstrating that in addition to quantitative changes in the LDL fraction, LA treatment also affects qualitative changes in this fraction [40,41]. In parallel, LA results in a higher ratio of newly produced LDL particles, which are more resistant to oxidative modifications [41], compared to older, partially modified lipoprotein particles, which may be one of the reasons for the lower susceptibility to lipoprotein oxidation after the LA procedure observed by us.

In the present work, in addition to the study of the direct effect of a single LA session, we also analyzed the long-term effect of LA by comparing the results of patients with high Lp(a) levels undergoing regular biweekly LA procedure treatment and patients with documented high Lp(a) levels who qualified for the LA procedure but did not consent to it. We observed a prolonged average value of oxidation lag time and a lower average value of the slope of the oxidation curve in the LA group than in the non-LA group for both Lp(a) and LDL fractions; however, the differences between the groups did not reach statistical significance (Table 5). Additionally, analysis of the kinetics of the formation of conjugated dienes in the Lp(a) and LDL fractions (Figure 4) may indicate a more beneficial course in patients undergoing regular LA, which prompts further research and seems to point to the potential additional long-term clinical effect of LA and the beneficial impact not only on the quantity but also on the quality of lipoprotein particles related to their susceptibility to oxidative modification. Our findings expand the knowledge of the pleiotropic effect of LA that may affect CVD risk. A prospective study showed that LA treatment causes an acute increase in anti-inflammatory fatty acid (FA) levels but also a reduction in FAs responsible for the development of CVD [38]. Another finding is that the LA can not only remove lipoproteins but also remove extracellular vesicles, and their removal may be associated with a reduction in CVD risk [39]. Additionally, regular LA induces a significantly improved endothelial function [42].

The proatherogenic mechanism of Lp(a) action is multidirectional and still poorly understood. Hence, it is not surprising that Lp(a) has attracted the interest of many clinical and scientific groups. An interesting aspect seems to also be the study of ATX. In the present work, we observed a significant reduction in the ATX activity immediately after the LA procedure (Table 3) as well as significantly higher baseline activity of ATX in patients not undergoing apheresis (Table 2). Additionally, we indicated that ATX activity positively correlated with Lp(a) concentrations in both groups (LA group: R = 0.48, *p* = 0.031; non-LA group: R = 0.55, *p* = 0.042). Our analysis of correlations between the parameters characterizing oxidative susceptibility of the Lp(a) fraction and the activity of ATX showed that there was a significant correlation between serum ATX activity and all analyzed parameters in patients undergoing and not undergoing the LA procedure (Table 4), which may suggest the importance of ATX for ASCVD risk. Our results confirm the significance of ATX activity in individuals with high Lp(a) concentrations as a concept worthy of attention and exploration.

Considering the strong association of Lp(a) and LDL-C concentrations with ASCVD risk, we also analyzed the correlations between parameters of oxidation susceptibility and the concentration of lipoproteins in the group of subjects undergoing and not undergoing LA treatment. As shown in Table 4, significant correlations between the concentration of Lp(a) and LDL-C and parameters describing oxidative susceptibility are in line with the fact that with high concentrations of proatherogenic lipoprotein, their quality is also modified with a higher probability of their oxidation. Raal et al., investigating the relationship between the quantity and quality of LDL particles in patients with familial hypercholesterolemia (FH), found that these patients had large, buoyant LDL particles, which are less susceptible to oxidation [43]. These results confirmed the reduction in proatherogenic lipoprotein concentration as the main therapeutic goal and, on the other hand, showed that the future of supportive antioxidant therapy should still be considered. In our work, the group analyzed comprised patients with hyperlipoproteinemia(a), and the obtained results demonstrated a decrease in the quality of lipoprotein particles, expressed as resistance to oxidation, as their concentration increases.

The LA procedure has demonstrated an acute decrease in Lp(a) and LDL-C concentrations associated with a consistent reduction in the occurrence of adverse cardiac or vascular events. As previous studies have shown, a single LA session lowers the concentration of Lp(a) and LDL-C by more than 60% [31,40,44]. The results of our work are in line with these studies and showed that the greatest reduction after LA using the MONET technique was in the concentration of Lp(a) and LDL-C, by 62% and 48%, respectively (Table 3). Our earlier study showed that a single LA treatment reduces oxidative stress markers, such as 8-iso-PGF2α, in a group of patients with FH [40]. However, we did not confirm this observation in our group of patients with isolated hyperlipoproteinemia(a), and no significant differences were found in the 8-iso-PGF2α concentration before and after LA. Oxidative stress-related parameters are still under investigation; however, the effects of the LA procedure on these parameters are rare and not clear. Some works have indicated that LA may decrease oxidative stress biomarkers [45,46], while others have demonstrated the strengthening of the oxidation process [47,48].

There are also inconclusive studies showing the effect of apheresis on the qualitative changes in particular classes of lipoproteins, including HDL, which may have a significant impact on lipoprotein metabolism and proatherogenic properties of all lipoproteins fractions [40,49]. We evaluated the apolipoprotein profile in the HDL fraction and showed a significant increase in HDL apoA-I/apoE and apoA-I/apoC-III ratios after the LA procedure (Table 3). This may suggest that LA preferentially removes HDL particles rich in apoE and apoC-III. Depending on the allele of the *APOE* genotype, some researchers show apoE-associated increased cardiovascular risk, while others show a protective effect in, for example, myocardial infarction [50]. The study by Morton et al. demonstrated that apolipoproteins E and C-III interact to regulate HDL metabolism, and apoC-III abolishes apoE’s benefit of reducing CHD risk [51]. Our analysis of the apolipoprotein profile of the HDL fraction showed that among analyzed apolipoproteins, LA has the greatest impact on apoC-III. Considering the fact that apoC-III seems to attenuate the antiatherogenic properties of HDL and has an effect on oxidative modifications of proatherogenic lipoproteins [52,53], the effect of apheresis on reducing apoC-III in HDL seems to be a particularly positive aspect in their quality associated with the CVD risk. No significant differences were observed in HDL apoA-I/apoE and apoA-I/apoC-III ratios between the LA group and non-LA group.

LA is considered a safe and well-tolerated procedure; however, the cyclical rebound of Lp(a) and LDL-C within 1 to 2 weeks between procedures makes it necessary for LA to take place every 1 or 2 weeks [54]. Recommendations for the administration of LA in patients with high Lp(a) levels are not clear and vary between countries. In 2008, the German Federal Joint Committee (GBA) approved weekly LA for subjects with isolated high Lp(a) levels (>60 mg/dL), with the concentration of LDL-C on target and progressive CVD despite effective treatment of all other CVD risk factors [40,54,55,56]. In Poland, LA treatment is usually carried out at 14-day intervals and is indicated for isolated hyperlipoproteinemia(a) with an Lp(a) level above 60 mg/dL and progressive CVD [31,34,35,38]. The analysis of the kinetics of the LA-induced changes seems to be important, so we assessed lipid and apolipoprotein profile parameters as parameters that also characterize susceptibility to oxidative modification of Lp(a) and LDL fractions 7 days after the LA procedure. No significant differences were observed between all analyzed parameters before and 7 days after apheresis. Due to the fact that levels of all measured parameters return to their baseline values 7 days after the LA session, our results may be an important element in the ongoing debate of whether the procedure should be performed at two- or one-week intervals.

A limitation of this study is the relatively small sample size. The sample size was partly a consequence of the number of patients who were under the care of our center and agreed to undergo regular biweekly LA procedures, the number of patients who agreed to participate in the studies, and the exclusion criteria that influenced the final number of the entire population. The small sample size may have contributed to the lack of statistical differences between parameters describing susceptibility to oxidative modification of Lp(a) and LDL fractions in patients undergoing and not undergoing LA procedures (Table 5).

## 4. Materials and Methods

### 4.1. Patients

The study groups included 43 patients with hyperlipoproteinemia(a), aged 31–82 years, from the First Clinic of Cardiology at the Medical University of Gdańsk (Poland). Among them, there were 24 individuals undergoing biweekly LA (LA group) and 19 individuals matched for lipoprotein(a) (Lp(a)) concentration and gender who qualified for but were undergoing LA treatment due to a lack of consent for this procedure (non-LA group). The indication for LA was hyperlipoproteinemia(a) with an Lp(a) level above 100 mg/dL, progressive ASCVDs, and LDL-C levels on target. Regular LA treatment was performed at biweekly intervals using the cascade filtration method (MONET), and the average treatment time was 3–4 h. The anticoagulation was based on heparin and citrate infusion. The LA procedure was designed and conducted to achieve Lp(a) reduction of at least 60% and processed at least 45 mL of plasma volume per kg of body weight.

Inclusion criteria were as follows: adults over 18 years of age, conscious and voluntary consent to participate in this study. The criteria for exclusion from this study were as follows: lack of informed consent for participation in this study, an active inflammatory process measured by C-reactive protein (CRP) (concentration > 10.0 mg/L) and morphology, untreated hypothyroidism, severe chronic kidney failure (glomerular filtration rate < 30 mL/min/1.73 m^2^), administration of anti-inflammatory or immunosuppressive drugs, and pregnancy. Clinical data regarding family history of CVDs, arterial hypertension, diabetes, chronic kidney disease, obesity, dyslipidemia, smoking status, and medications taken were collected via a questionnaire completed by the patient. According to the American Heart Association, CVDs were considered to be a group of disorders of the heart and blood vessels and included heart disease, heart attack, stroke, heart failure, arrhythmia, and also heart valve problems. This study was performed in accordance with the ethical guidelines of the 1975 Declaration of Helsinki and was approved by the Medical Ethics Committee of the Medical University of Gdańsk (approval number KB/428/2018-2022). All of the participants provided written informed consent.

### 4.2. Sample Collection

Blood samples from the patients undergoing biweekly LA procedures were obtained from peripheral blood in a fasting state at three time points: directly before the LA session, immediately after the procedure, and 7 days after the procedure (between two LA sessions). Blood samples from subjects with documented high Lp(a) levels but who were not treated with LA were collected during a single scheduled meeting. The serum was separated after centrifugation at 1000× *g* for 15 min, and then part of the material was immediately ultracentrifuged in a density gradient to obtain lipoprotein fractions. The rest of the serum, as well as the isolated lipoprotein fractions, were stored at −80 °C until analysis.

### 4.3. Isolation of LDL and Lp(a) Fractions

The ultracentrifugation-based method was used to isolate appropriate lipoprotein fractions from fresh serum samples. Sequential density gradient ultracentrifugation was performed using an Optima TLX-120 ultracentrifuge equipped with a TLA-120.2 angle rotor (Beckman Coulter Inc., Boulevard Brea, CA, USA). Isolation was carried out in three stages at a centrifugal force of 100,000× *g* for 2.5, 5, and 3 h, respectively, at 16 °C. In the first stage, 775 μL serum aliquot was pipetted into an 11 × 25 mm Quick-Seal Polypropylene Ultracentrifuge Tube (Beckman Coulter, CA, USA). A discontinuous gradient was formed by carefully layering 775 μL of 0.9% NaCl on top of the serum, followed by sealing the tube with a soldering iron. After 2.5 h of ultracentrifugation, the upper fraction containing very low-density lipoprotein (VLDL: <1.006 g/mL) was collected, and 775 µL of the lower fraction was transferred to the next ultracentrifuge tube with the layered 16.7% NaCl in a 1:1 ratio. The tubes were sealed and subjected to ultracentrifugation for 5 h. After the process, the upper fraction containing low-density lipoprotein (LDL: 1.006–1.063 g/mL) was collected. Finally, 775 µL of the lower fraction obtained from the second step was transferred to the next ultracentrifuge tube and layered with 15.2% NaCl in a 1:1 ratio. After 3 h of ultracentrifugation, an upper fraction containing lipoprotein(a) (Lp(a): 1.063–1.087 g/mL) and a lower fraction containing high-density lipoprotein (HDL: 1.063–1.210 g/mL) were obtained. The purity of the isolated blood serum fractions was confirmed by agarose gel electrophoresis (Sebia, Lisses, France), and undetectable levels of apoA-I in the Lp(a) fraction and apo(a) in the HDL fraction measured by the methods used in the project.

### 4.4. Laboratory Measurements

Total cholesterol (TC), low-density lipoprotein cholesterol (LDL-C), and triglycerides (TG) were measured using standard enzymatic colorimetric tests (Wiener Laboratorios SAIC, Buenos Aires, Argentina). The high-density lipoprotein (HDL) was isolated by the precipitation of apolipoprotein B-containing lipoproteins with heparin and manganese chloride, and the HDL-cholesterol (HDL-C) was determined in the supernatant using a colorimetric test from Wiener Lab. The concentrations of apolipoproteins apoA-I, apoB, apoC-III, and apoE were measured using the nephelometric method with antibodies obtained from Siemens Healthcare Diagnostics (Eschborn, Germany) on a Behring laser nephelometer. Total protein was determined with a biuret reagent with Lowry’s modification. The lipoprotein(a) (Lp(a)) concentration was performed using a commercially available immunoturbidimetric assay with a Five-Point Calibrator, which provides a reduction in the size differences of apo(a) isoforms (Randox, Crumlin, UK). Autotaxin (ATX) activity in serum was analyzed based on a hydrolysis rate of 1 myristoyl-sn-glycero-3-phosphocholine by measuring choline as a reaction product by colorimetric assay (Merck, Warsaw, Poland). The activity unit of ATX was defined as 1 µmol of 1 myristoyl-sn-glycero-3-phosphocholine formed per minute per liter of serum at 37 °C. 8-isoprostane PGF2α (8-iso-PGF2α) levels in serum were measured by enzyme immunoassay kits (Cayman Chemical, Ann Arbor, MI, USA).

### 4.5. Copper-Mediated Oxidation of LDL and Lp(a) Fractions

Assessment of the oxidation susceptibility of LDL and Lp(a) fractions was carried out in the presence of Cu^2+^ ions. This resulted in the formation of conjugated dienes—products of the oxidation reaction of lipoprotein particles. The formation of conjugated dienes was measured by kinetic absorbance readings. Briefly, the lipoprotein fraction was adjusted to 0.1 mg/mL protein. Copper sulfate was added to a final concentration of 5 µM. After thorough mixing, the samples were transferred to a 96-well plate, and the absorbance readings at λ = 234 nm at 37 °C were taken every 5 min by a spectrophotometer (Thermo Scientific, Waltham, MA, USA).

A phase of linear increase in absorbance (R > 0.99) was determined for the absorbance of each sample. The preceding phase was defined as the lag phase (oxidation lag time). The slope angle of the oxidation curve in the linear phase was calculated based on the directional coefficient of the equation of the straight line, which is the tangent of the slope angle of the curve with linear scaling of the OX and OY axes. We defined the increase in absorbance as the maximum change in the difference between the initial absorbance and the maximum absorbance achieved (Figure 5).

### 4.6. Statistical Analysis

Statistical analyses were performed using STATISTICA software version 10 (StatSoft, Warsaw, Poland) and GraphPad Prism version 4.0 (GraphPad Software, San Diego, CA, USA). A Shapiro–Wilk test was used to assess the normality of the distribution of variables. The variables were expressed as mean ± SD (standard deviation) or as medians with 25th and 75th percentiles. Data analysis was performed using the Student’s *t*-test and nonparametric Mann–Whitney U test. Pearson’s chi-square test was used to compare categorical variables, and Spearman’s standardized coefficients were used to assess univariate correlations. The changes in individual parameters due to LA sessions (after and 7 days after LA) were analyzed using a parametric or non-parametric ANOVA test for dependent variables and Dunn’s or Holm–Sidak’s post hoc multiple comparisons analysis. The statistical significance of differences between the mean absorbances in LA and non-LA groups at all time points was assessed using the Wilcoxon test. A *p*-value < 0.05 was considered to be statistically significant.

## 5. Conclusions

Our data demonstrated that the LA procedure has an impact on the quality of Lp(a) and LDL particles related to their reduced oxidative susceptibility and is connected with Lp(a) ATX activity. Considering the clinical relevance of Lp(a) and LDL for ASCVDs and the importance of oxidized forms of lipoproteins in the pathogenesis of atherosclerosis, changes in the lipoprotein susceptibility to oxidation in patients undergoing LA may have a significant impact on their proatherogenic properties and may represent the pleiotropic effect of LA associated with a reduction in the rate of CV events. Additionally, our results showing a rebound effect 7 days after LA will contribute to a better determination of frequency guidelines for LA sessions for patients with high concentrations of Lp(a) and will make grounds for improvement of a therapeutic strategy.

## Figures and Tables

**Figure 1 ijms-25-13597-f001:**
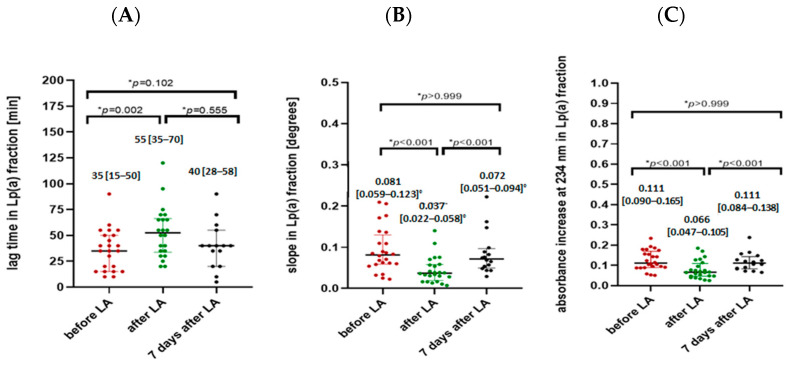
Lag time (**A**), the slope of the oxidation curve (**B**), and the increase in absorbance at 234 nm (**C**) in Lp(a) fraction in patients before LA, after LA, and 7 days after LA procedure. Data are presented as median [25–75 percentile range]. * Non-parametric ANOVA test for dependent variables and Dunn’s post hoc multiple comparisons analysis. °-degrees.

**Figure 2 ijms-25-13597-f002:**
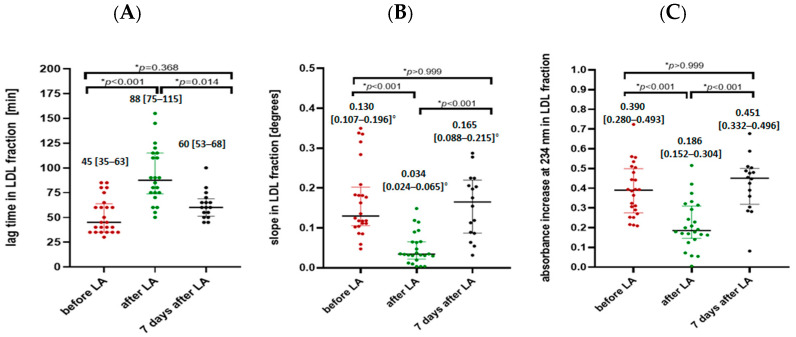
Lag time (**A**), the slope of the oxidation curve (**B**), and the increase in absorbance at 234 nm (**C**) in LDL fraction in patients before LA, after LA, and 7 days after LA procedure. Data are presented as median [25–75 percentile range]. * non-parametric ANOVA test for dependent variables and Dunn’s post-hoc multiple comparisons analysis. °-degrees.

**Figure 3 ijms-25-13597-f003:**
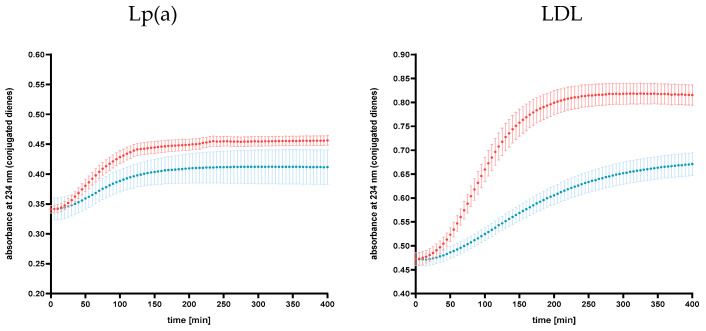
Comparison of oxidation curves of Lp(a) fraction and LDL fraction in patients undergoing biweekly LA procedure (before LA session: red, immediately after LA session: blue). Points are presented as mean absorbance ± standard deviation of the mean at each time point.The statistical significance of differences between the means at all time points was assessed using the Wilcoxon test.

**Figure 4 ijms-25-13597-f004:**
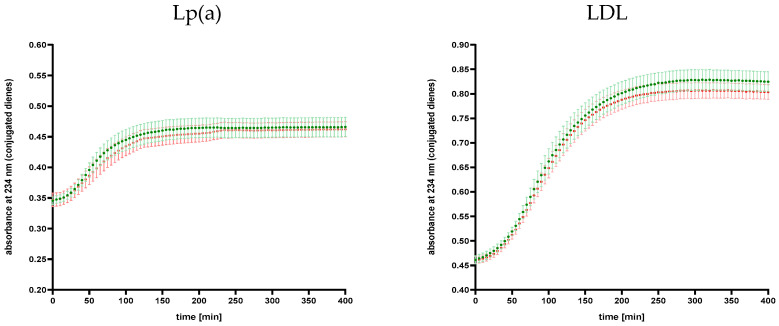
Comparison of oxidation curves of Lp(a) fraction and LDL fraction in patients undergoing (LA group: red) and not undergoing (non-LA group: green) LA procedure. In the LA group, the values before LA procedure were analyzed. Points are presented as mean absorbance ± standard deviation of the mean at each time point.The statistical significance of differences between the means at all time points was assessed using the Wilcoxon test.

**Figure 5 ijms-25-13597-f005:**
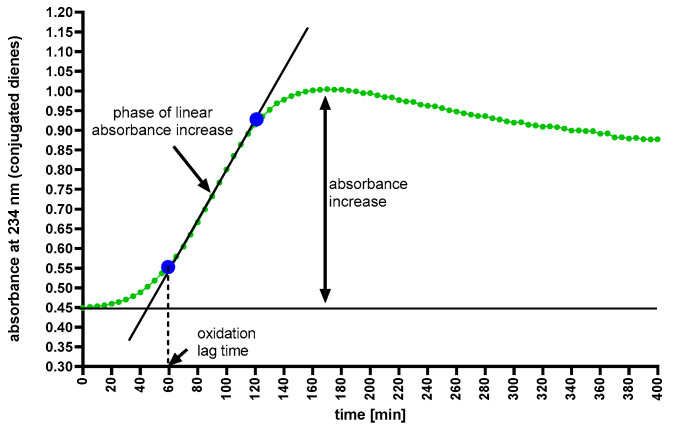
Example of an oxidation curve for the low-density lipoprotein (LDL) fraction and parameters for assessing the fraction’s susceptibility to oxidation.

**Table 1 ijms-25-13597-t001:** Baseline clinical characteristics of the study groups.

	LA Group N = 24	Non-LA Group N = 19	*p*
Sex (female–male), N	5:19	6:13	0.423 **
Age, years	58 ± 12	68 ± 8	0.004 *
Hypertension, N (%)	14 (58%)	12 (63%)	0.485 **
Diabetes, N (%)	4 (17%)	4 (21%)	0.596 **
Smoking habits, N (%)	11 (46%)	8 (42%)	0.282 **

Age values are presented as mean ± standard deviation. Data analysis was performed using * Student’s *t*-test and ** Pearson’s chi-square test. LA group—patients undergoing lipoprotein apheresis (LA); non-LA group—patients not undergoing LA.

**Table 2 ijms-25-13597-t002:** Biochemical characteristics.

	LA Group N = 24	Non-LA Group N = 19	*p*
Lp(a), mg/dL	140 (126–167)	123 (110–149)	0.085 **
TC, mg/dL	124 (103–157)	148 (135–162)	0.104 **
LDL-C, mg/dL	60 (46–69)	63 (46–89)	0.723 **
HDL-C, mg/dL	48 (42–64)	61 (48–73)	0.142 **
TG, mg/dL	75 (65–128)	90 (69–122)	0.385 **
Phospholipids, mg/dL	176 ± 47	186 ± 28	0.417 *
ATX, [IU/L]	2.29 (1.76–3.14)	3.09 (2.15–3.93)	0.013 **
8-iso-PGF2α, pg/mL	10.1 (7.9–15.8)	10.6 (7.7–12.7)	0.578 **
serum apoA-I, mg/dL	149 (136–175)	176 (154–206)	0.106 **
serum apoB, mg/dL	70.5 (57.2–92.9)	74.5 (65.2–80.8)	0.667 **
serum apoB/apoA-I	0.46 (0.40–0.62)	0.43 (0.35–0.52)	0.464 **
serum apoC-III, mg/dL	4.23 (3.73–6.04)	4.87 (3.57–5.86)	0.912 **
serum apoE, mg/dL	3.13 ± 1.28	4.16 ± 0.84	0.005 *
HDL apoA-I, mg/dL	108 (90–128)	109 (103–122)	0.779 **
HDL apoE, mg/dL	0.65 (0.48–0.91)	0.65 (0.48–0.86)	0.687 **
HDL apoA-I/apoE	162 (118–228)	167 (124–253)	0.669 **
HDL apoC-III, mg/dL	0.72 (0.40–1.14)	0.59 (0.40–0.68)	0.279 **
HDL apoA-I/apoC-III	148 (102–210)	196 (157–225)	0.062 **

In the LA group, the values of parameters before LA procedure were analyzed. Values are presented as mean ± standard deviation or as median (25th and 75th percentiles). Data analysis was performed using * Student’s *t*-test and ** nonparametric Mann–Whitney U test. Abbreviations: Lp(a)—lipoprotein(a); TC—total cholesterol; LDL-C—low-density lipoprotein cholesterol; HDL-C—high-density lipoprotein cholesterol; TG—triglycerides; apoA-I-apolipoprotein A-I; apoB—apolipoprotein B; apoC-II—apolipoprotein C-II; apoC-III—apolipoprotein C-III; apoE—apolipoprotein E; apoB/apoA-I —apoB to apoA-I ratio; ATX—autotaxin; 8-iso-PGF2α —8-iso-prostaglandin F2aα; HDL apoA-I—apoA-I in HDL fraction; HDL apoE—apoE in HDL fraction; HDL apoA-I/apoE—apoA-I to apoE ratio in HDL fraction; HDL apoC-III—apoC-III in HDL fraction; HDL apoA-I/apoC-III—apoA-I to apoC-III ratio in HDL fraction.

**Table 3 ijms-25-13597-t003:** Impact of single LA procedure on biochemical serum parameters and apolipoprotein profile in the HDL fraction.

Parameters	Before LA	After LA	*p*	7 Days After LA	# *p*
TC, mg/dL	124 (103–157)	73 (51–85)	<0.001 *	141 (105–161)	>0.999 *
LDL-C, mg/dL	60 (46–69)	31 (22–37)	<0.001 *	58 (45–78)	0.368 *
HDL-C, mg/dL	48 (42–64)	42 (36–52)	0.002 *	50 (45–59)	0.648 *
TG, mg/dL	75 (65–128)	41 (30–52)	<0.001 *	92 (75–131)	>0.999 *
Lp(a), mg/dL	140 (126–167)	53 (45–66)	<0.001 *	121 (106–183)	>0.999 *
Phospholipids, mg/dL	176 ± 47	114 ± 28	<0.001 **	181 ± 44	0.638 **
ATX, [IU/L]	2.29 (1.76–3.14)	1.98 (1.56–2.64)	0.011 *	2.36 (2.09–3.09)	>0.999 *
8-iso-PGF2α, pg/mL	10.1 (7.9–15.8)	10.0 (8.3–11.5)	0.119 *	9.7 (6.4–10.8)	0.119 *
serum apoA-I, mg/dL	149 (136–175)	121 (108–150)	0.003 *	151 (129–208)	0.259 *
serum apoB, mg/dL	70.5 (57.2–92.9)	34.8 (23.6–49.6)	<0.001 *	74.5 (51.0–90.8)	>0.999 *
apoB/apoA-I	0.46 (0.40–0.62)	0.26 (0.23–0.31)	<0.001 *	0.41 (0.38–0.57)	0.910 *
serum apoC-III, mg/dL	4.23 (3.73–6.04)	2.23 (1.63–2.90)	<0.001 *	4.51 (3.85–5.73)	>0.999 *
serum apoE, mg/dL	3.13 ± 1.28	2.02 ± 0.82	<0.001 **	3.21 ± 1.09	0.675 **
HDL apoA-I, mg/dL	108 (90–128)	97 (77–105)	<0.001 *	112 (97–151)	>0.999 *
HDL apoE, mg/dL	0.65 (0.48–0.91)	0.49 (0.26–0.64)	<0.001 *	0.71 (0.44–1.19)	>0.999 *
HDL apoA-I/apoE	162 (118–228)	189 (157–407)	<0.001 *	166 (119–252)	>0.999 *
HDL apoC-III, mg/dL	0.72 (0.40–1.14)	0.23 (0.17–0.34)	<0.001 *	0.73 (0.51–1.02)	0.662 *
HDL apoA-I/apoC-III	148 (102–210)	363 (272–451)	<0.001 *	131 (107–176)	>0.999 *

Values are presented as mean ± standard deviation or as median (25th and 75th percentiles). # *p*—*p*-value for differences before and 7 days after LA procedure. Differences between the results were analyzed using a parametric or non-parametric ANOVA test for dependent variables and * Dunn’s or ** Holm-Sidak’s post hoc multiple comparisons analysis.

**Table 4 ijms-25-13597-t004:** Correlation analysis between parameters characterizing susceptibility to oxidative modification: in the Lp(a) fraction—serum Lp(a) concentration and ATX activity; in the LDL fraction—serum LDL-C concentration, in patients undergoing and not undergoing LA procedure.

Lp(a) or LDL Fraction Parameter	Lp(a) Concentration				ATX Activity				LDL-C Concentration			
	LA Group		Non-LA Group		LA Group		Non-LA Group		LA Group		Non-LA Group	
	R	*p*	R	*p*	R	*p*	R	*p*	R	*p*	R	*p*
oxidation lag time	−0.43	0.062	−0.55	0.044	−0.46	0.038	−0.54	0.022	−0.30	0.153	−0.35	0.142
slope of the oxidation curve	0.47	0.021	0.13	0.604	0.48	0.028	0.56	0.013	0.59	0.002	0.44	0.060
increase in absorbance at 234 nm	0.39	0.062	0.12	0.624	0.55	0.010	0.56	0.013	0.26	0.214	0.24	0.322

In the LA group, the values before LA procedure were analyzed. Correlation analysis was performed using the Spearman rank correlation test.

**Table 5 ijms-25-13597-t005:** Comparison of parameters describing susceptibility to oxidative modification of Lp(a) and LDL fractions in patients undergoing and not undergoing LA procedure.

Parameters	Lp(a) Fraction			LDL Fraction		
	LA Group	Non-LA Group	*p*	LA Group	Non-LA Group	*p*
oxidation lag time [min]	35 [15–50]	30 [25–40]	0.789	45 [35–63]	35 [35–70]	0.404
slope of the oxidation curve [degrees]	0.081 [0.059–0.123]	0.110 [0.045–0.150]	0.576	0.130 [0.107–0.196]	0.165 [0.121–0.224]	0.417
increase in absorbance at 234 nm	0.111 [0.090–0.165]	0.133 [0.094–0.186]	0.533	0.390 [0.280–0.493]	0.377 [0.337–0.463]	0.620

In the LA group, the values before LA procedure were analyzed. Values are presented as median (25th and 75th percentiles). Data analysis was performed using the non-parametric Mann–Whitney U test.

## Data Availability

The data presented in this study are available upon request from the corresponding author.

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
