# Peer review of "Reduced Oxidative Susceptibility of Lp(a) and LDL Fractions as a Pleiotropic Effect of Lipoprotein Apheresis in Patients with Elevated Lp(a) and ASCVDs"

_ijms, 2024, doi:10.3390/ijms252413597_

Round 1
Reviewer 1 Report
Comments and Suggestions for Authors
In this manuscript, Krzesinska et al investigated the oxidative modifications of lipoproteins in the plasma of patients who undergo apheresis. The study compared the lipoprotein fractions, autotaxin activity and conjugated dienes in samples from patients who qualified for apheresis and patients who refused apheresis. Also, the samples from patients who performed apheresis were examined at three different time points, before, immediately after apheresis and after 7 days from apheresis.
The manuscript is well written, however, I have some remaining concerns:
1. In line 51 it is an additional space.
2. Emphasize the importance of the study for the medical field when describe the aim.
3. What is the purpose of measuring the blood parameters 7 days after the apheresis procedure given that the procedure is biweekly? For the measured parameters in Table 3, a single apheresis procedure is mentioned. However, in the Methods section, the sample collection was done from patients with biweekly procedures. Refrain for a better understanding.
4. For the results presented in Figure 3 mention if the parameters were measured after 7 days (for a single procedure) or in samples collected from patients who undergo biweekly procedures. And mention at what time point after the procedure the blood was collected in the case of the biweekly procedure.
5. A comparison of the oxidation curves of Lp(a) and LDL fractions between non-apheresis and apheresis groups after 7 days would be interesting.
6. In line 361 from the Discussion section revise the English formulation.
Author Response
Thank you very much for reviewing our work and for all valid suggestions to improve it. Please, consider our point by point response to all comments while we have modified the revised version of the manuscript accordingly.
Reviewer 1. Comments and Suggestions for Authors
In this manuscript, Krzesinska et al investigated the oxidative modifications of lipoproteins in the plasma of patients who undergo apheresis. The study compared the lipoprotein fractions, autotaxin activity and conjugated dienes in samples from patients who qualified for apheresis and patients who refused apheresis. Also, the samples from patients who performed apheresis were examined at three different time points, before, immediately after apheresis and after 7 days from apheresis.
The manuscript is well written, however, I have some remaining concerns:
Comment 1. In line 51 it is an additional space.
Response 1: We thank the Reviewer for this coment and we apologize for this error. We have removed the extra space in line 51 and we have reviewed the manuscript once again.
Comment 2. Emphasize the importance of the study for the medical field when describe the aim.
Response 2: We are grateful to the Reviewer for this valuable comment and we apologize for not highlighting this important aspect. We have completed the aim in the Introduction section.
“The aim of our study was to assess the effects of LA procedure on quality of lipopro-tein fractions in patients with high Lp(a) concentrations undergoing regular biweekly LA treatment. Additionally, we analyzed the quality of lipoprotein fractions in patients who were qualified for LA treatment but do not consent to this procedure. Considering the clinical importance of oxidized forms of lipoproteins in the pathogenesis of atherosclerosis, our study may also help to understand the pleiotropic effect of LA associated with a reduction in CV events. In addition, the results 7 days after LA may be the basis to better define guidelines for the frequency of LA use in patients with high Lp(a) concentrations.”
Comment 3. What is the purpose of measuring the blood parameters 7 days after the apheresis procedure given that the procedure is biweekly? For the measured parameters in Table 3, a single apheresis procedure is mentioned. However, in the Methods section, the sample collection was done from patients with biweekly procedures. Refrain for a better understanding.
Response 3: We thank the Reviewer for this question. The measurement of blood parameters 7 days after apheresis aims to understand the dynamics of changes in biochemical parameters, particularly proatherogenic Lp(a) particles as well as lipid profile and apolipoprotein profile. The results of the measurements 7 days after the apheresis procedure showed that the studied parameters are returning to their pre-procedure values. These rebound effect may provide a basis for increasing the frequency of apheresis - from every two weeks (currently performed) to every week.
In the Methods section,we described that samples were taken from patients undergoing biweekly procedures, although the same patients, at our request, also came to the clinic 7 days after LA for blood collection. Therefore, it was possible to assess the effect of a single apheresis treatment on the blood parameters listed in Table 3.
Comment 4. For the results presented in Figure 3 mention if the parameters were measured after 7 days (for a single procedure) or in samples collected from patients who undergo biweekly procedures. And mention at what time point after the procedure the blood was collected in the case of the biweekly procedure.
Response 4: We are grateful to the Reviewer for this important comment. In line with the Reviewer's suggestions, we have made appropriate corrections to the description of Figure 3.
Comment 5. A comparison of the oxidation curves of Lp(a) and LDL fractions between non-apheresis and apheresis groups after 7 days would be interesting.
Response 5: We thank the Reviewer for this valid suggestion. We agree with the fact that the additional analysis would be interesting, however, our goal was to assess whether regularly performed apheresis could have a long-term effect on the reduction of baseline values of oxidative susceptibility parameters in patients undergoing apheresis procedure every two weeks. Therefore, we compared the values before apheresis with the values in patients not undergoing apheresis. In addition, we did not observe significant differences in any of the parameters assessing lipoprotein oxidative susceptibility in patients before and 7 days after LA treatment, both in the Lp(a) and LDL fractions, therefore plotting dates before and 7 days after LA together a graph makes it unreadable.
Comment 6. In line 361 from the Discussion section revise the English formulation.
Response 6: We thank the Reviewer for this relevant comment. We have corrected the formulation.
“German Federal Joint Committee (GBA) approved weekly LA for subjects with isolated high Lp(a) levels (>60 mg/dl), concentration of LDL-C on target and progressive CVD despite effective treatment of all other CVD risk factors.”
Reviewer 2 Report
Comments and Suggestions for Authors
The work presented is of particular interest and well explained in the manuscript. The scientific approach is mainly based on the analysis of the contribution of oxidative state and this certainly adds important information to the study of atherosclerotic condition.
Does the choice of population reflect the real epidemiology of atherosclerosis or CVD?
The discussion of data is very clear but also excessively long, you risk losing focus, I recommend to the authors to rearrange it, considering the possibility of putting some info in the introductory section or summarize it.
Author Response
Thank you very much for your review. We very appreciate the important and valid comments. We hope that we have managed to respond to all the comments in a manner that the reviewer will find satisfactory.
Reviewer 2. Comments and Suggestions for Authors
The work presented is of particular interest and well explained in the manuscript. The scientific approach is mainly based on the analysis of the contribution of oxidative state and this certainly adds important information to the study of atherosclerotic condition.
Comment 1. Does the choice of population reflect the real epidemiology of atherosclerosis or CVD?
Response 1: We thank the Reviewer for this question. Data from the Copenhagen General Population Study revealing that elevated Lp(a) level >50 mg/dL is present in 20% of general population and an awareness of the strong association between CVD risk and high Lp(a) concentration sets new goals in treatments for patients with ASCVD. High Lp(a) concentration is strictly associated with an increased risk of stroke, myocardial infarction, peripheral artery disease, heart failure, as also cardiovascular mortality. Therefore, we selected a group of patients with hyperlipoproteinaemia(a) who have progressive ASCVD and are treated with apheresis procedure.
Comment 2. The discussion of data is very clear but also excessively long, you risk losing focus, I recommend to the authors to rearrange it, considering the possibility of putting some info in the introductory section or summarize it.
Response 2: We are grateful for drawing our attention to this important aspect and for valid suggestions. The length of the discussion was due to the number of performed analyses, as also the desire to discuss in detail the results obtained. Following the Reviewer's advice, we have moved the section on autotaxin to the introductory section and removed the description of some of the results, which are included in Table 3.